# Reinforcement learning with Demonstrations from Mismatched Task under Sparse Reward

**Yanjiang Guo**[1], **Jingyue Gao**[1], **Zheng Wu**[2], **Chengming Shi**[1], **Jianyu Chen**[1,3,*]
[1]Institute for Interdisciplinary Information Sciences, Tsinghua University
[2]University of California, Berkeley
[3]Shanghai Qizhi Institute
{guoyj22, gaojy19, shicm19}@mails.tsinghua.edu.cn, zheng_wu@berkeley.edu
* Correspondence to: Jianyu Chen (jianyuchen@tsinghua.edu.cn)

**Abstract:** Reinforcement learning often suffer from the sparse reward issue in real-world robotics problems. Learning from demonstration (LfD) is an effective way to eliminate this problem, which leverages collected expert data to aid online learning. Prior works often assume that the learning agent and the expert aim to accomplish the same task, which requires collecting new data for every new task. In this paper, we consider the case where the target task is mismatched from but similar with that of the expert. Such setting can be challenging and we found existing LfD methods can not effectively guide learning in mismatched new tasks with sparse rewards. We propose conservative reward shaping from demonstration (CRSfD), which shapes the sparse rewards using estimated expert value function. To accelerate learning processes, CRSfD guides the agent to conservatively explore around demonstrations. Experimental results of robot manipulation tasks show that our approach outperforms baseline LfD methods when transferring demonstrations collected in a single task to other different but similar tasks.

**Keywords:** Sparse Reward Reinforcement Learning, Learn from Demonstration, Task Mismatch

## 1 Introduction

Reinforcement learning has been applied to various real-world tasks, including robotic manipulation with large state-action spaces and sparse reward signals [1]. In these tasks, standard reinforcement learning tends to perform a lot of useless exploration and easily fall into local optimal solutions. To eliminate this problem, previous works often use expert demonstrations to aid online learning, which adopt some successful trajectories to guide the exploration process [2, 3].

However, standard learning from demonstration algorithms often assume that the target leaning task is exactly same with the task where demonstrations are collected [4, 5, 6]. Under this assumption, experts need to collect the corresponding demonstration for each new task, which can be expensive and inefficient. In this paper, we consider a new learning setting where expert data is collected under a single task, while the agent is required to solve different new tasks. For instance as shown in Figure 1, a robot arm aims to solve peg-in-hole tasks.The demonstration is collected on a certain type of hole while the target tasks have different hole shapes (changes in environmental dynamics) or position shifts (changes in reward function). This can be challenging as agents cannot directly imitate those demonstrations from mismatched tasks due to dynamics and reward function changes. However, compared to learning from scratch, those demonstrations should still be able to provide some useful information to help exploration.

6th Conference on Robot Learning (CoRL 2022), Auckland, New Zealand.

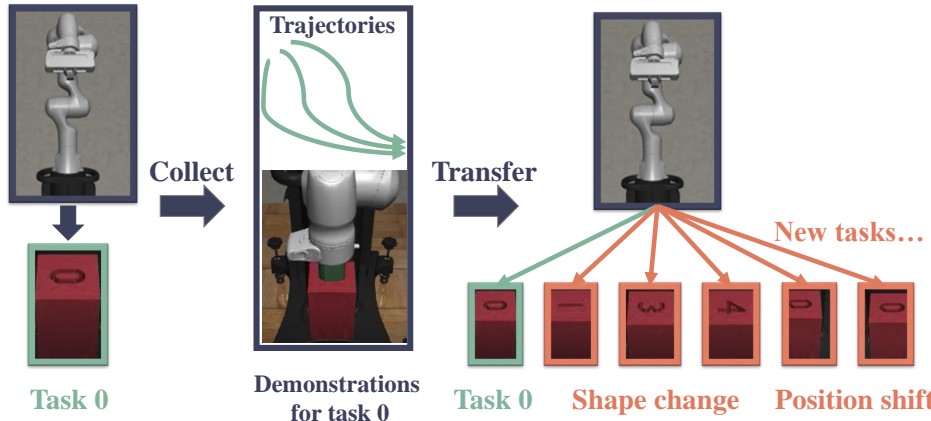

Figure 1: Illustration of our motivation. Demonstrations collected on a single original task are transferred to other similar but different tasks with either environmental dynamics changes (shape change) or reward function change (position shift), and aid the learning of these tasks.

To address the issue of learning with demonstrations from mismatched task, previous works in imitation learning consider agent dynamics mismatch and rely on state-only demonstrations [7, 8, 9]. However, this approach has an implicit assumption that the new task share the same reward function as the original task [10]. Hester et al. and Vecerik et al. [11, 3] receive sparse rewards in the environment and add demos into a prioritized replay buffer. Sparse reward signal can be backward propagated during the Bellman update and thus guide the exploration. However, this propagation flow may be blocked due to the mismatch in new tasks. Another class of work [12, 13, 14] also considers that we have expert data on multitasks and utilize meta-learning methods to obtain diverse skills, and then transfer skills to new tasks. However, such a strategy requires to collect a huge expert dataset, which is expensive and inefficient. In our setting, we are only provided with a few demonstrations collected under a single task.

In this paper, we propose **C**onservative **R**eward **S**haping **f**rom **D**emonstration (**CRSfD**), which learns policies for new tasks accelerated by demonstrations collected in a single mismatched task. We use reward shaping [15, 16] to incorporate future information into single-step rewards while keeping the optimal policy unchanged. Moreover, we explicitly deal with out-of-distribution problem to encourage agent to explore around demonstrations. Experimental results of robot manipulation tasks show that our approach outperforms baseline LfD methods when learning in new tasks with mismatched demonstrations.

Our contributions can be summarized as follows:

- We proposed a reward shaping scheme for reinforcement learning with demonstration from mismatched task, which use estimated value function from expert demonstrations to re-shape sparse reward in new tasks.
- Built upon such scheme, we propose the conservative reward shaping from demonstration (CRSfD) algorithm to overcome the out-of-distribution problem, we regress value function of OOD states to zero and use a larger discount factor in new tasks, which guides the agent to conservatively explore around expert data.
- We conduct simulation and real world experiments of robot insertion tasks with mismatched demonstrations. The results show that CRSfD effectively guide the exploration process in new tasks and reach a higher sample efficiency and convergence performance.

## 2   Related Works

**Learning from demonstration** A prominent research subject is how to leverage expert data to assist reinforcement learning. Imitation learning (IL) is a broad family of such algorithms that enforce agents to directly imitate the expert. Behavior cloning (BC) is the simplest IL algorithm which

greedily imitates the step-wise action of the expert and can fall into the problem of distributional shift [17]. Inverse reinforcement learning [18, 4] and adversarial imitation learning [6] infer the expert's reward function and learn the corresponding optimal policy jointly. The above IL algorithms assume environment rewards are not available, hence their performances are upper-bounded by that of experts [19]. Another line of work makes use of reward feedback from environment and leverages expert demonstration data to overcome the sparse reward issue or learn more natural behaviors. Vecerik et al. [3] add demonstration into a prioritized replay buffer. Rajeswaran et al. [20] add a behavioral cloning loss to the policy to speed up exploration and learn more natural and robust behaviors. Chen et al. [21] use generative models on single step transition to reshape reward of the original task. However, standard learning from demonstration algorithms always requires demonstrations to be collected under the same task and act nearly optimal under this task, which is not suitable for our setting.

**Generalization of demonstrations** There are a few works relaxing the requirements for demonstrations to achieve generalization of demonstrations from different aspects. Some works assume that demonstrations are collected by a sub-optimal policy under the same task [22, 23], early work [23] requires manually ranking of trajectories and later works [24, 25] move the needs for rankings by actively adding noise to demonstrations along with automatical ranking. Cao et al. [26, 27] assume that the demonstrations are a mixture of different experts and use a classifier to separate out the more feasible expert data for the new task. Other works [10, 28, 29] assume that the target task has different agent dynamics to the task where demonstrations are collected, so they only match the state sequence of demonstrations or use an inverse dynamic model to recover the action between two states in the new task. In our work, we further consider new tasks with the environment dynamics mismatch as well as reward function mismatch. Another branch of related works are meta imitation learning algorithms, which assume that we have expert data on multitasks and utilize meta-learning methods to solve new tasks in zero-shot or few shots adaption [12, 13]. However, such a strategy usually necessitates a huge expert dataset which may be expensive and inefficient. Differently, we consider the problem where only a small number of demonstrations collected in a single task are provided, and the agent needs to use them to accelerate the learning of other similar but different tasks.

# 3 Problem Statement

In our problem setting, we have collected a few demonstrations under a single task and want to utilize these data in reinforcement learning for other similar but different tasks. A task can be formalized as a standard Markov decision process MDP, which is modeled as $M_i = (S, A, P_i, R_i, \gamma_i)$. The task where demonstrations are collected is denoted as $M_0 = (S, A, P_0, R_0, \gamma_0)$, and the new tasks we target to solve are denoted as $M_i = (S, A, P_i, R_i, \gamma_i), i \geq 1$. $S$ and $A$ are the shared state space and action space for each task. $P_i : S \times A \times S \to [0, 1]$ are state transition probability functions of each task, $R_i : S \times A \times S \to \mathbb{R}$ are reward functions for task $M_i$, describing the natural reward signal in each task. Due to differences of environment and agent dynamics, $P_i$ and $R_i$ often varied between different tasks. $\gamma_i$ is discounted factor of $M_i$ which reflects how much we care about future, typically set to a constant slightly lower than 1. A policy $\pi_i : S \to A$ defines a probability distribution in action space. For a task $M_i$ and a policy $\pi_i$, state value function $V_i^{\pi_i}(s) = \mathbb{E}_{s_0=s, \pi_i}[\Sigma \gamma_i^t R_t(s_t, a, s_{t+1})]$ estimates the discounted cumulative reward of the task under this policy $\pi_i$. $V_i^*(s)$ estimates the discounted cumulative rewards for state $s$ under the optimal policy $\pi_i$.

As many works [30, 31] point out, directly applying RL in a sparse reward environment can be sample inefficient and fail to find a good solution. In this work, we want to make use of the demonstrations $D : (\tau_0, \tau_1, ...)$ collected in task $M_0$ to facilitate reinforcement learning for the different but similar new tasks $M_i$. Note each trajectory $\tau_k$ contains a sequence of state action transitions $[s_0, a_0, s_1, a_1, ... s_t, a_t]$ in task $M_0$.

**Challenges** There are two key issues when leveraging demonstrations from mismatched tasks. First, how to get effective guidance from these mismatched demonstrations? Although we should not purely imitate these demonstrations, we do need to obtain some useful guidance from them to acceleration exploration in new tasks with sparse rewards. Second, since our goal is to maximize the reward defined under the new task, guidance from mismatched demonstrations should not influence the optimality of the learned policy in new tasks.

# 4 Conservative Reward Shaping from Demonstrations

Provided with demonstrations in a particular task $M_0 : (S, A, P_0, R_0, \gamma_0)$, we aim to help the reinforcement learning process of different tasks $M_1, M_2...M_K$, which may have different transition functions $P_k(s'|s,a)$ and reward functions $R_k(s,a,s')$. In this work, we use SAC [32] as our base reinforcement learning algorithm as it holds an excellent exploration mechanism which leads to higher sample efficiency than policy gradient algorithms [33, 34] and is shown to perform well on continuous action tasks [32]. Nevertheless, it is also possible to base our method on other RL algorithms including on-policy ones. To make use of expert demonstrations, DDPGfD [3] proposes a mechanism compatible with the off-policy method, which adds the demonstration data into the replay buffer with prioritized sampling. Under such framework, the sparse reward signal can propagate back along the expert trajectory to guide the agent. By combining SAC and DDPGfD [3], we obtain the backbone of our method and labeled as SACfD, which is also our best baseline method.

## 4.1 Reward Conflict under Mismatched Task Setting

Although LfD methods such as SACfD benefit from demonstrations in sparse reward reinforcement learning, they may not benefit from demonstrations when the target tasks are mismatched from that of the expert. When following the demonstrations, agent may consistently fail and can not get any sparse rewards signals. As failure time increases, agent may consider expert trajectories to have low value since few rewards are received. The agent will then avoid following the expert and the demonstrations cannot provide effective guidance, resulting in inefficient exploration in the whole free space.

Although totally following the demonstrations may not be able to receive any sparse reward in new tasks, it can still provide useful exploration directions since in our settings the new tasks are similar to the original one. We formally introduce our method as conservative reward shaping from demonstration (CRSfD). Intuitively, CRSfD assigns appropriate reward signals along the demonstrations to efficiently guide the agent towards the goal, and allows exploration around the goal to maintain optimally. Details are described in the following subsection.

## 4.2 Conservative Reward Shaping from Demonstrations(CRSfD)

**Reward Shaping with Value Function** Reward shaping [15] provides an elegant way to modify reward function while keeping the optimal policy unchanged. Given original MDP $M$ and an arbitrary potential function $\Phi : S \rightarrow \mathbb{R}$, we can reshape the reward function to be:

$$R'(s,a,s') = R(s,a,s') + \gamma\Phi(s') - \Phi(s), s' \sim P(.|s,a) \tag{1}$$

Denote the new MDP as $M' = (S, A, P, R', \gamma)$ obtained by replacing reward function $R$ in $M$ to $R'$. Ng et al. [15] proved that the optimal policy $\pi_{M'}^*$ on $M'$ and the optimal policy $\pi_M^*$ on the original MDP $M_0$ are the same: $\pi_{M'}^* = \pi_M^*$. Furthermore, the optimal state-action function $Q_{M'}^*(s,a)$ and value function $V_{M'}^*(s)$ are shifted by $\Phi(s)$:

$$Q_{M'}^*(s,a) = Q_M^*(s,a) - \Phi(s), \quad V_{M'}^*(s) = V_M^*(s) - \Phi(s) \tag{2}$$

In particular, Ng et al. [15] pointed out that when the potential function is chosen as the optimal value function of the original MDP $\Phi(s) = V_M^*(s)$, then the new MDP $M'$ becomes trivial to solve. What remained for the agent is to choose each time-step's action greedily, because the transformed single-step reward already contains all the long-term information for decision making.

**Conservative Value Function Estimation** The reward shaping method provides a principled way to guide the agent with useful future information and keep the optimal policy unchanged. Ideally, an accurate $\Phi_i(s) = V_{M_i}^*(s)$ will lead to simple and optimal policy in new MDP $M'$, but a perfect $\Phi_i(s) = V_{M_i}^*(s)$ is unavailable in advance. Practically, we estimate a $\widetilde{V}_{M_0}^D \approx V_{M_0}^*(s)$ using demonstrations from task $M_0$ by Monte-Carlo regression and treat $\widetilde{V}_{M_0}^D(s)$ as a prior guess of $V_{M_i}^*(s)$. We then shape the sparse reward in the new task $M_i$ to:

$$R_i'(s,a,s') = R_i(s,a,s') + \gamma\widetilde{V}_{M_0}^D(s') - \widetilde{V}_{M_0}^D(s) \tag{3}$$

However, demonstration trajectories only cover a small part of the state space. For out-of-distribution states, estimated $\widetilde{V}_{M_0}^D$ may output random values and lead to random single-step reward after reward shaping, which may mislead the agent. We make two improvements over the above reward shaping method to encourage the agent to explore around the demonstrations conser-

vatively: (1) Regress value function $\widetilde{V}^*_{M_0}(s)$ of the out-of-distribution states to 0, thus discouraging exploration far from demonstrations. The OOD states are sampled randomly from free space. (2) Increasing the discount factor $\gamma_i$ in new tasks. From equation 3, we can find that increasing $\gamma_i$ will give higher single-step reward for state with large $V_\theta(s')$ in the original task, thus encourages exploration around demonstrations. Our method can be summarized as follows: ($D$ stands for demonstration buffer, $S$ stands for free space, $\gamma_i > \gamma_0$):

---

**Algorithm 1** Conservative Value Function Estimation

---

**Input:** Demonstration transitions, demo discount factor $\gamma_0$, new task discount factor $\gamma_k(\gamma_k > \gamma_0)$, regression steps $n_r$, scale factor $\lambda$.

**Initialization:** Initialize value function $V_\theta(s)$

Monte-Carlo policy evaluation on demonstrations, Calculate cumulative reward for states in demos using $\gamma_0$: $V^D_{M_0}(s) = \Sigma^T_{i=t}\gamma_0^{i-t}r_i$

**for** $n$ in regression steps $n_r$ **do**

    Sample minibatch $B_1$ from demo buffer $D$ with regression target $V^D_{M_0}(s) = \Sigma^T_{i=t}\gamma_0^{i-t}r_i$. Sample minibatch $B_2$ from whole free space $S$ with regression target 0.

    perform regression: $\theta = \arg\min\limits_{\theta} \left[ \mathbb{E}_{s_t\sim B_1}\left(V_\theta(s_t) - \Sigma^T_{i=t}\gamma_0^{i-t}r_i\right)^2 + \lambda\mathbb{E}_{s_t\sim B_2}\left(V_\theta(s_t) - 0\right)^2 \right]$

**end for**

Shaping reward with $\gamma_k$: $R'_i(s,a,s') = R_i(s,a,s') + \gamma_kV_\theta(s') - V_\theta(s)$.

Perform SACfD update. (detalis can be found in appendix.)

---

**Conservative Properties** In the last paragraph, we introduced some conservative techniques and give some intuitively explanations why those improvements can encourage exploration around demonstrations under the proposed reward shaping framework. The following theorem can quantize the benefits of proposed methods.

**Theorem 1** *For task $M_0$ with transition $T_0$ and new task $M_k$ with transition $T_k$, define total variation divergence $D_{TV}(s,a) = \Sigma_{s'}|T_0(s'|s,a) - T_k(s'|s,a)| = \delta$. If we have $\delta < (\gamma_k - \gamma_0)\mathbb{E}_{T_2(s'|s,a)}[V^D_{M_0}(s')]/\gamma_0 \max_{s'} V^D_{M_0}(s')$, then following the expert policy in new task will result in immediate reward greater then 0:*

$$\mathbb{E}_{a\sim\pi(.|s)}r'(s,a) \geq (\gamma_k - \gamma_0)\mathbb{E}_{T_k(s'|s)}[V^D_{M_0}(s')] - \gamma_0\delta\max_{s'}V^D_{M_0}(s') > 0 \qquad (4)$$

Detailed proof can be found in Appendix 7.5. The above theorem indicates that for similar but different tasks ($\delta$ smaller than the threshold), exploration along demonstrations will lead to positive immediate rewards which guide the learning process.

**Conservative Reward Shaping from Demonstrations** After reward shaping by demonstrations from mismatched task, we perform online learning based on SACfD as described in Section 4. Pseudocode can be found in supplementary materials. Although the estimated $\widetilde{V}^D_{M_0}(s)$ can be inaccurate, it still provides enough future information, thus facilitates exploration for the agent. Moreover, nice theoretical properties of reward shaping guarantees that we will not introduce bias to the learned policy in new tasks.

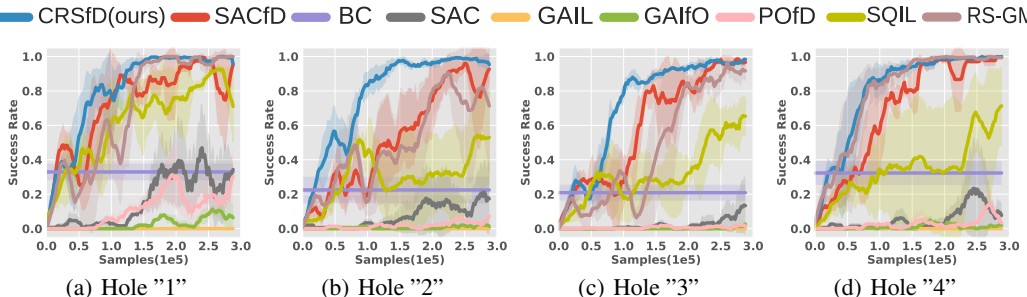

Figure 2: Evaluation of algorithms on 4 new tasks with demonstrations from task "0". The solid line corresponds to the mean of success rate over 5 random seeds and the shaded region corresponds to the standard deviation. Y-axis reflects success rate range in [0, 1], X-axis reflects interaction steps range in [0, 3e5].

# 5 Experimental Results

We perform experimental evaluations of the proposed CRSfD method and try to answer the following two questions: Can CRSfD help the exploration of similar sparse-rewarded tasks with demonstrations from a mismatched task? Will CRSfD introduce bias to the learned policy in new tasks?

We choose the robot insertion tasks for our experiments, which has natural sparse reward signals: successfully inserting peg into hole get a reward +1, otherwise 0. We perform both simulation and real world experiments. The simulation environment is built under robosuite framework [35] powered by Mujoco physics simulator [36]. We construct a series of similar tasks where the holes have different shapes and unknown position shifts, reflecting changes in dynamics and reward functions respectively, as shown in Figure 1. Then we verify the effectiveness of CRSfD under the following 2 settings: (1) Transfer collected demonstrations to similar insertion tasks with environment dynamics mismatch. (2) Transfer collected demonstrations to similar tasks with both environment dynamics and reward function mismatch. Finally, we address the sim-to-real issue and deploy the learned policy on a real robot arm to perform insertion tasks with various shapes of holes in the real world. We use Franka Panda robot arm in both simulation and real world. The comparison baseline algorithms are chosen as follows:

- **Behavior Cloning [17]**: Just ignore the task mismatch and directly perform behavior cloning of the demonstrations.
- **SAC [32]**: A SOTA standard RL method which does not use the demonstrations and directly learn from scratch in the target tasks.
- **GAIL [6]**: Use adversarial training to recover the policy that generates the demonstrations, which allieviates the distributional shift problem of behavior cloning.
- **GAIfO [8]**: A variant of GAIL which trains a discriminator with state transitions $(s, s')$ instead of $(s, a)$ in GAIL to alleviate dynamics mismatch.
- **POfD [37]**: A variant of GAIL which combines the intrinsic reward from discriminator and extrinsic reward from the new task.
- **SQIL [38]**: An effective off-policy imitation learning algorithm that adds demonstrations with reward +1 to the buffer and assign reward 0 to all agent experiences.
- **SACfD [32, 3]**: Incorporate effective demonstration replay mechanism from [3] with SAC as described in Section 4, which is also the best baseline as well as the backbone of our method.
- **RS-GM [21]**: Reward Shaping using Generative Models, which is an extension on discrete reward shaping methods [39, 40]. After learning a discriminator $D_\phi(s, a)$, they shape the reward into $R'(s, a) = R(s, a) + \gamma\lambda D_\phi(s', a') - \lambda D_\phi(s, a)$.

## 5.1 Simulation experiments

We set a nominal hole position as the original-point of our Cartesian coordination. Observable states include robot proprioceptive information such as joint and end-effector position and velocity. Action space includes the 6d pose change of the robot end-effector in 10 Hz, followed with a Cartesian impedance PD controller running at a high frequency. Only a sparse reward +1 is provided when the peg is totally inserted inside the hole. Demonstrations are collected by a sub-optimal RL policy trained with SAC in task $M_0$ under carefully designed dense reward, where the hole has shape "0". This process can be replaced by manual collection in the real world. Then demonstrations are tagged with the corresponding sparse reward. We collected 40 demonstration trails with 50 time steps each.

**Setting 1: Tasks with environment dynamics mismatch.** To reflect environment dynamics changes of the tasks, we create experiments domain on insertion tasks with holes of various shapes in the simulator, represented by different digit numbers, as shown in Figure 1. Different shapes of holes will encounter different contact mode thus lead to different environmental dynamics. We collect demonstrations from hole "0", and our method use them to help training similar new tasks with various hole shapes from digit 1 to 4.

**Analysis 1:** The comparison results of CRSfD and baseline algorithms under the above setting are show in Figure 2. As we expected, the simplest BC algorithm simply imitates the expert action of

the original task and can only complete the insertion with a small chance. The SAC algorithm does not make use of the demonstration data and conducts a lot of useless exploration, which leads to poor performance. GAIL algorithm and its variants GAIfO, POfD also fail for most of times as they try to purely imitate the demonstration collected in the mismatched task. SQIL ignores the reward in the new task and only obtains a limited success rate. SACfD can not be effectively guided by demonstrations from the mismatched task under sparse reward. Our proposed CRSfD provide guidance through reward shaping, and consistently achieves the best performance on all the four insertion tasks with different hole shapes.

**Setting 2: Tasks with both dynamics and reward function mismatch** Next, we consider more challenging scenarios where we aim to transfer the demonstrations to new tasks with both environmental dynamics mismatch and reward function mismatch. We assume that the hole has unknown random shifts relative to the nominal position, thus the reward function changes. At the beginning of each episode, the hole is uniformly initialized in a square area centered at the nominal position. This can be challenging because the robot is 'blind' to these unknown offsets and requires further search for the entrance of the hole. Practically, we collected demonstrations from task with hole "0" with fixed hole position, and transfer to new tasks with random hole shifts and different hole shapes.

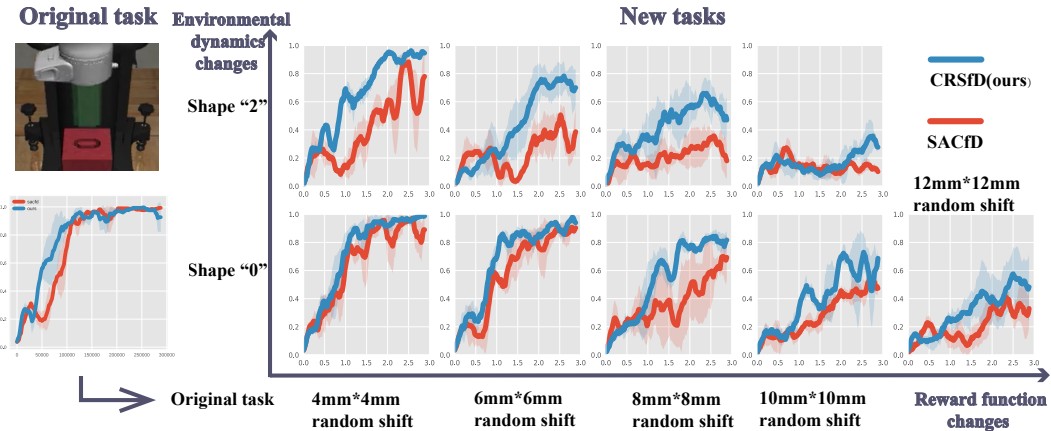

Figure 3: Evaluations of CRSfD and the best baseline SACfD. The solid line corresponds to the mean of success rate over 3 random seeds and the shaded region corresponds to the standard deviation. X-coordinate reflects changes in reward functions and Y-coordinate reflects changes in environmental dynamics. Our algorithm outperforms baseline with increasing margins as the task changes become larger.

**Analysis 2:** We compare our algorithm with the best baseline algorithm SACfD under varying degrees of environmental dynamics and reward function changes, as shown in the Figure 3. Due to space limit, more comparison can be found in Figure **??** in appendix. The x-coordinate represents the increasing changes of the reward function, where the random range of the holes becomes larger (from 4mm*4mm, 6mm*6mm, to 8mm*8mm). The y-coordinate represents increasing environmental dynamics change, from hole "0" in its original shape to hole "2" in a different shape. Straightforwardly, coordinate origin can represent the original task where demonstrations are collected, and a 2d coordinates $[x, y]$ represents a new task with varying degree of mismatch.

From Figure 3, we can observe that when applying to the original task or very similar task such as [4mm, shape '0'], our method has a similar performance to the SACfD baseline. When the task changes become greater (e.g, [8mm, shape '0'], [4mm, shape '2'], [6mm, shape '2'], [8mm, shape '2']), SACfD gradually lose the guidance from original demonstrations as task mismatched more significantly, while CRSfD achieves significant performance gains with help of the conservative reward shaping using estimated value function.

**Ablation study** As mentioned in section 4.2, we make two improvements over the reward shaping method to encourage the agent to explore around the demonstrations conservatively. (1) Regress value function of OOD states to zero. (2) Use a larger discount factor in new tasks.

We ablate these 2 improvements and compare their performance. Ablations are tested under new task with hole shape "3", results for other shapes can be found in the supplementary materials. As shown in Figure 4, compared to original CRSfD algorithm, moving away either of these 2 techniques will lead to a performance drop, where the agent needs to take more effort in exploration.

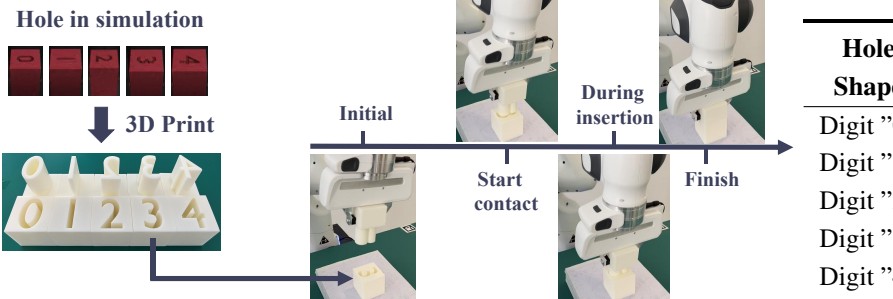

Figure 4: Ablation studies of the conservativeness techniques. (1) means regressing value function to zero for OOD states. (2) means setting larger discount factors.

### 5.2 Real World Experiments

After completing the insertion tasks of various-shaped holes in the simulator, we deploy the policy to the real robotic arm. To overcome the sim-to-real problem, we use domain randomization in the simulation. The initial position of the robot arm end-effector and holes are randomized in a 6cm*6cm*6cm space and 2mm*2mm plane respectively, and the friction coefficient of the object is also randomized in [1, 2]. We use a real Franka Panda robot arm and 3d print the holes corresponding to digit numbers "0-4". Holes are roughly in sizes of 4cm*4cm, with a 1mm clearance between the peg and the hole. We performed 25 insertion trials under each shape of hole, and counted their success rates separately, as shown in the table 1. The robot achieves high success rate in all tasks.

Figure 5: Real world robot insertion experiments.

| Hole Shape | Success Rate |
|---|---|
| Digit "0" | 1.0 |
| Digit "1" | 1.0 |
| Digit "2" | 0.92 |
| Digit "3" | 0.92 |
| Digit "4" | 0.96 |

Table 1: Success rate for real world robot insertion tasks.

## 6 Conclusion

**Summary.** In this paper, we studied the problem of reinforcement learning with demonstrations from mismatched tasks under sparse rewards. Our key insight is that, although we should not purely imitate the mismatched demonstrations, we can still get useful guidance from the demonstrations collected in a similar task. Concretely, we proposed conservative reward shaping from demonstrations (CRSfD) which uses reward shaping by estimated value function of a mismatched expert to incorporate useful future information to augment the sparse reward, with conservativeness techniques to handle out-of-distribution issues. Simulation and real world robot insertion experiments show the effective of proposed method under tasks varied in environmental dynamics and reward functions.

**Limitations and Future works.** Provided with demonstrations from a mismatched task, our proposed method aids the online learning process for each new task separately. However, one may need to learn a policy to solve multiple new tasks at the same time, and exploration in these tasks may benefit each other. So future works include using demonstrations to accelerate the joint learning process of multiple tasks. Another limitation is that our method is only applicable to new tasks similar to original task. The effectiveness of CRSfD gradually decays when the tasks differ too much from the original task so that the demonstrations do not contain any useful information. It also worth to mention that the whole algorithm pipeline should be able to be implemented directly on hardware, which is a promising research direction.

**Acknowledgments**

This work is supported by the Ministry of Science and Technology of the People's Republic of China, the 2030 Innovation Megaprojects "Program on New Generation Artificial Intelligence" (Grant No. 2021AAA0150000).

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
