# OpenReview forum: "Reinforcement learning with Demonstrations from Mismatched Task under Sparse Reward"
_robot-learning.org/CoRL/2022/Conference — CoRL 2022 Poster_

### Official Review · Reviewer_Mmxm · 2022-07-26

**Originality:** Fair
**Technical Quality:** Good
**Clarity Of Presentation:** Good
**Impact:** 3

**Recommendation:**

Weak Accept: I recommend accepting the paper, but will not argue for my recommendation if the majority of other reviewers have a different opinion.

**Summary:**

The paper studies how demonstrations can be used to improve the performance of online reinforcement learning (RL). In particular, the authors study settings where the demonstrated task does not exactly match the tasks which robot will have to execute. This removes the requirement of obtaining expert demonstrations for every new task the robot encounters. The main contribution is a conservative reward shaping algorithm for reinforcement learning with demonstrations from mismatched tasks. The work is evaluated in simulation and real-world experiments, including comparisons against state-of-the-art Learning from Demonstration (LfD) methods.

**Issues:**

1) Resolved -- Add a theoretical discussion on how the proposed algorithm avoids Sparse Reward Signal Backward Propagation Blockage
2) Resolved -- Elaborate the description of the regression and discount factor (lines 176-180), as well as the conservative character of the approach.
3) Resolved -- Extend evaluation to at least one of the following: i) increasingly larger task mismatch, ii) entirely different problem (plots for ii could just be in the appendix).

**Quality Of The Limitations Section:**

Additional details required

**Reviewer Expertise:**

3: The reviewer is fairly confident that the evaluation is correct

**Robotics Focus:**

Sufficient demonstration on hardware

**Strengths And Weaknesses:**

Overall, the paper is well written, clearly structured and addresses an important challenge in robot learning – using demonstrations from mismatched tasks in RL. The authors identify shortcomings of existing approaches, and propose a novel solution technique. The evaluation in simulation and real-world experiments deliver convincing results.
On the downside, the paper lacks detail in the theoretical part, and the evaluation is limited to only one experiment. Please see detailed comments below:

•	resolved -- The paper identifies a key issue for standard methods combining LfD with RL, called Sparse Reward Signal Backward Propagation Blockage. While the description and illustration are convincing on a high level, I would like to see more technical details on this.

•	resolved -- Related to above comment, the paper lacks a more rigorous description of how the use of reward shaping avoids the backward propagation blockage. Are there any theoretical guarantees on this?

•	resolved -- The limited theoretical discussion also raises questions about the overall contribution of the paper: is the use of reward shaping with demonstrations novel? I don’t think any related work for this was cited, yet it is not entirely clear to me.

•	Updated -- The second main contribution is addressing the out-of-distribution problem. The approach for this is only described very briefly in lines 176-180, and leaves questions about the details: How does the regression work? How is the discount factor increased?

Update: While the overall description has improved in the revised paper, the details about how the discount factor is updated are still unclear (including the extended algorithm in the appendix)

•	resolved -- What is the ‘Conservative’ part of the proposed technique? Is there a less or more conservative variant of the algorithm, and how does it compare? Since it appears in the name of the algorithm, it should be elaborated more.

•	Updated -- The parameter \lambda in equation 3 is not explained.

Update: Similar to the discount factor, I am still missing details on how to choose this p.arameter

•	Updated -- While the evaluation is overall providing detailed insights and delivers promising results, the major weakness is that it only considers slight variations of a peg-into-hole problem. It could be significantly strengthened by being extended to a completely different robot task.

Update: the paper improved greatly by additional experiments with varying mismatch. Thus, the experiments are rich enough for a conference paper.

•	resolved -- An important question left unanswered is how mismatched the tasks can be. While it might be difficult to get specific theoretical results, empirical data would be interesting. I would expect that with increasing mismatch, the advantages of the proposed approach will start to disappear. For large mismatch, using information from the demonstrations might actually be detrimental to learning progress.


Minor comments:
•	resolved -- Typo line 77 Standard should be lower cas: standard

•	resolved -- Line 156, shouldn’t the function map be  $\Phi : S \rightarrow \mathbb{R}$ not … \rightarrow R ?

Limitation Section:
•	resolved -- The limitation section only discusses joint learning for multiple tasks. While this is an interesting future work direction, it is only partially a limitation of the presented work – multiple tasks is not part of the problem statement. Some discussion of how to find the discount factor, if there are problems where the proposed method is not beneficial at all, or the influence how mismatched demonstrations are would be very interesting



**Summary Of Recommendation:**

The paper discusses an interesting and relevant problem and is well structured. However, the technical presentation lacks some more detailed description and discussion and thus is not entirely convincing.

Update: The technical description has improved through the revisions, and the experiments were extended which helped to show the robustness of the proposed work.

---

> ### Author Response · Authors · 2022-08-22
> **Response to Reviewer Mmxm - Part(1/2)**
>
> Thank you so much for your insightful comments on our paper. Your review helps us a lot on revising our paper. We have uploaded the revised paper with modified places marked in blue, and we hope our following responses answer your concerns about our paper.
>
> * **About the technical details of reward signal propagation issue and theoretical analysis.**
>
> To make the description more clear, we have simplified the explanation of blocked reward signal issue in section 4.1 and removed Figure 2. You can intuitively understand it as following: imitating the original demonstrations in new tasks may lead to consistent failure and obtain no sparse rewards from the environment. Our method uses reward shaping to incorporate the information from the demonstrations to the sparse reward of the new task, instead of purely imitating the original demonstrations. By doing so, we can guide the learning process while keeping the policy optimal in the new task.
> In addition, we also provide detailed description of the algorithm and some theoretical analysis (in Section 4.2) to help better understanding. You can find details of the modifications (marked blue) in the revised paper.  Thank you for your time!
>
> * **Is the use of reward shaping with demonstrations novel? I don’t think any related work for this was cited, yet it is not entirely clear to me.**
>
> Indeed, using reward shaping for a given sparse reward task is not a new idea. The novelty of the paper is to look in particular at domain shift between the target task and the demonstration task. This is a very important problem in robotic learning with demonstrations, since robotic tasks are usually pretty diverse (different shapes and positions of the holes for peg-in-hole tasks as considered in this paper), and collecting demonstrations for all tasks can be extremely expensive and inefficient. Therefore, the goal of this paper is to learn better policies for a variety of target tasks, when only demonstrations from a single different task are provided. This will allow us to handle new tasks better and faster without the need of collecting new demonstration data. To do this, we use reward shaping to incorporate the information from the demonstrations to the sparse reward of the new task, which accelerates the training process of new tasks while keeping the optimal policy unchanged. This is novel among approaches of learning with demonstrations from mismatched tasks, and performances better than baselines.
>
> * **Elaborate the description of the regression and discount factor (lines 176-180), as well as the conservative character of the approach.**
>
> Thanks for your suggestion! We have revised the description in the paper to provide some more detailed descriptions and pseudo-code (marked blue in section 4.2 of the revised paper). Our method estimates a state-value function using expert data from the original task. However, the demonstrations only cover a little part of states in state space so the state value function may output inaccurate value in the states not covered by the demonstrations, which leads to suboptimal reward shaping. Therefore, we perform regularization on the value function by randomly sampling states from free space and regress their value to 0. By doing so, we encourage the agent to "conservatively" explore around the demonstrations, avoiding inefficient random explorations.
> Naturally, doing so would introduce bias into the state-value function, but the elegant theoretical guarantees of the reward shaping with potential function ensures that this bias will not affect the optimal policy. When the reward is shaped, the discount factor is the coefficient before the state value function in Equation (3) of the revised paper. Increasing the discount factor can make the high-value states around the expert data have higher single-step rewards which further encourage exploration around the demonstrations.

---

> > ### Author Response · Authors · 2022-08-22
> > **Response to Reviewer Mmxm - Part(2/2)**
> >
> >
> > * **An important question left unanswered is how mismatched the tasks can be. While it might be difficult to get specific theoretical results, empirical data would be interesting. I would expect that with increasing mismatch, the advantages of the proposed approach will start to disappear.**
> >
> > This is an insightful comment! Let us consider two extreme cases, if the new task is exactly the same as the old task, there exist many mature LfD algorithms. Taking the other extreme case, if the new task is totally different from the old task, the demonstrations of the old task should not help the new task at all. Our method attempts to fill the gap between the two extreme, using transferred demonstrations to solve tasks that have some similarities while not totally the same.  To better explore how mismatched the tasks can be, we also extend our experiments to more significant shift situations. We gradually increase the hole random range and compare the performance between our method and the strongest baseline. Results are shown in Figure 3 of the revised paper and Figure 7 in revised appendix. Our proposed methods outperforms baseline until the task largely shift with 14mm*14mm random position.
> > Thanks again for your comments that helped us further refine the experiment.
> >
> > * **Typos in line 77, 156. The parameter $\lambda$ in equation 3 is not explained**
> >
> > Thank you so much! We have fixed all the typos you mentioned in revised paper and added explanation of $\lambda$ in Algorithm 1.
> >
> > * **Limitation section:Some discussion of how to find the discount factor, if there are problems where the proposed method is not beneficial at all, or the influence how mismatched demonstrations are would be very interesting**
> >
> > We add Theorem 1 which provides a theoretical analysis of how the discount factor influence the shaped rewards. However, practically we may not be able to obtain a precise value.
> > We have also updated our limitation section in revised paper (marked in blue).
> > Our methods can be effective only in new tasks which have similarities to original tasks so that original demonstrations can provide useful information.
> >
> >
> >
> > Thanks again for your time and insightful comments!

---

> > ### Comment · Reviewer_Mmxm · 2022-08-27
> > **response to authors**
> >
> > I appreciate the authors' efforts into revising the paper. My concerns have mostly be addressed, I in particular the technical description and the additional experiment with various mismatch.
> > However, I have some more minor comments:
> >
> > Lines 253-273, setting 2 and analysis 2.In the entire description you only refer to Figure 7 from the appendix, Fig 3 from the main paper is not referred to at all. Line 250 and 271 mention the reward signal propagation blockage problem, but in the revision this term had been removed from the technical description and thus is confusing here.

---

> > > ### Author Response · Authors · 2022-08-28
> > > **Thanks so much for your kind and detailed comments and sorry for the confusion.**
> > >
> > > Thanks so much for your kind and detailed comments and sorry for the confusion. We revised the paper again in revised paper-v2 to refer both Fig.3 and Fig.7. At the same time, we also removed the description of the reward backward propagation blockage problem in paper line 250 and 271 for consistency. For your convenience, newly added changes are marked in red in revised paper-v2.
> > >
> > > We really appreciate your time in reading our revised papers, your constructive suggestions help us a lot in revising our paper! We hope we have resolved all the concerns and showed the improved quality of the paper. Thanks for your efforts and time again!

---

### Official Review · Reviewer_hEn1 · 2022-08-01

**Originality:** Fair
**Technical Quality:** Good
**Clarity Of Presentation:** Fair
**Impact:** 3

**Recommendation:**

Weak Accept: I recommend accepting the paper, but will not argue for my recommendation if the majority of other reviewers have a different opinion.

**Summary:**

Learning from Demonstration (LfD) is a mechanism to help guide reinforcement learning (RL) algorithms when in the setting of learning with sparse rewards. This paper explores a problem with using LfD to guide RL when the demonstrators providing the LfD data are performing a different ("mismatched") task. The paper develops Conservative Reward Shaping from Demonstration (CRSfD), which estimates and provides a shaped reward to the RL agent to support intelligent exploration. Empirical results show that the approach outperforms baselines, and the paper provides a robot demonstration.

**Issues:**

Please address the issues of related work, clarity in the presentation of the algorithm, and improved discussion/exploration of limitations.

**Quality Of The Limitations Section:**

Additional details required

**Reviewer Expertise:**

5: The reviewer is absolutely certain that the evaluation is correct and very familiar with the relevant literature

**Robotics Focus:**

Sufficient demonstration on hardware

**Strengths And Weaknesses:**

Strengths:
-The paper evaluates the proposed approach, CRSfD against numerous baselines, including both pure RL methods (SAC), pure LfD mechanisms (BC, GAIL, GAIfO), and hybrid appraoches (POfD, SQIL, and SACfD).
-The evaluation shows consistently positive results for CRSfD (Figure 3) across a set of custom tasks. The evaluation shows CRSfD can outperform its most challenging competitor, SACfD, when multiple aspects of the task have changed.
-An ablation study is provided to show the relative contributions of various aspects of the algorithmic framework for CRSfD.
-A real robot demonstration is provided (Table 1).
-Code is provided, along with a video.

Weaknesses:
-It would be helpful if the paper contained more details about the experimental domain (it wasn't until watching the video that it was clear what was going on) and why these tasks are the right tasks for benchmarking.
-This paper evaluates two forms of task misspecification (i.e., the (1) target location and (2) geometry for an insertion task move). However,
-The paper states that [22] "enforce[s] the agents to imitate the demonstrations only initially and then gradually discard them as learning progresses." However, in reading [22], it appears the method is grounded in the Preference-based Reinforcement Learning (PbRL) literature and generally does not imitate the demonstrations and gradually discard them. Rather, it takes a fundamentally different approach in which a reward function is learned based upon ranking data from humans, and a policy is optimized against that reward function. Later work by Brown et al. removed the need for rankings (Brown et al., CoRL'20), and Chen et al. then improved upon this method (CoRL'21). I think it would be helpful to benchmark SACfD against these approaches for learning from suboptimal demonstration.

Brown, D.S., Goo, W. and Niekum, S., 2020, May. Better-than-demonstrator imitation learning via automatically-ranked demonstrations. In Conference on robot learning (pp. 330-359). PMLR.

Chen, L., Paleja, R. and Gombolay, M., 2021, October. Learning from Suboptimal Demonstration via Self-Supervised Reward Regression. In Conference on Robot Learning (pp. 1262-1277). PMLR.

-Further limitations could have been discussed other than multi-task learning. What are the limits of the approach? When would it fail? Would this work for pick-and-place or other robot tasks? How does the approach degrade with the quality of the demonstration?
-The paper provides multiple figures (e.g., Figure 1 and Figure 2) that attempt to give intuition to the problem being studied. However, the paper spends very little time actually describing how the approach works (Lines 165-196). Pseudocode is relegated to the appendix. It would have been more helpful had the paper removed Figure 2 and spent more time describing the algorithm itself. The mismatched reward problem is relatively self-explanatory, and the "reward signal backward propagation blockage" doesn't seem to be a novel problem or even a problem that needs to be discussed at length. If a robot fails to accomplish a new task because it has learned a different one, it may not be able to receive any (sparse) rewards, so learning will be difficult. It isn't clear to this reviewer than anything beyond this point is being argued. Is the reviewer missing something? Also, is "reward backward propagation" the same issue than (though with different words -- perhaps a special case of) the canonical credit-assignment problem?
-Prior work has also considered reward shaping that should be considered:

Goyal, P., Niekum, S. and Mooney, R.J., 2020. Pixl2r: Guiding reinforcement learning using natural language by mapping pixels to rewards. arXiv preprint arXiv:2007.15543.

Questions:
-What would happen if one collected demonstrations on '4' but trained on '0' (or any other digit)?
-How could multi-task RL/IRL be helpful here? Consider:

Gleave, A. and Habryka, O., 2018. Multi-task maximum entropy inverse reinforcement learning. arXiv preprint arXiv:1805.08882.

Singh, A., Jang, E., Irpan, A., Kappler, D., Dalal, M., Levinev, S., Khansari, M. and Finn, C., 2020, May. Scalable multi-task imitation learning with autonomous improvement. In 2020 IEEE International Conference on Robotics and Automation (ICRA) (pp. 2167-2173). IEEE.

**Summary Of Recommendation:**

This paper develops an interesting reward-shaping approach to enable an RL algorithm to benefit from demonstrations that were provided for a different task. The results are positive; code and a robot demo are provided. However, the paper leaves out some important baselines (e.g., Chen et al., CoRL'21), does not devote enough attention to describing the technical method, and the limits of the algorithm are not sufficiently discussed/explored.

---

> ### Author Response · Authors · 2022-08-22
> **Response to Reviewer hEn1**
>
> Thank you so much for your insightful comments on our paper. Your review helps us a lot on revising our paper. We have uploaded the revised paper with modified places marked in blue, and we hope our following responses answer your concerns about our paper.
>
> * **It would be helpful if the paper contained more details about the experimental domain.**
>
> Thank you for your useful comments! We updated the description of our insertion domain in setting 1 of Section 5.1 in the revised paper.
>
> * **About description of literature [22]**
>
> Sorry for the confusing statement in the original paper. We have updated the description of the related papers in our revised paper.
>
> * **I think it would be helpful to benchmark SACfD against these approaches for learning from suboptimal demonstration.**
>
> Thanks for your insightful comments! However the demonstrations for these methods are collected in the **same task** with the target task, although they can be suboptimal. On the other hand, our paper targets on a different setting where the demonstrations are collected from a **different task**. Therefore, it might be unsuitable to compare them. Furthermore, both paper Brown et al., CoRL'20 and Chen et al. CoRL'21 do not release their code, and it is difficult to reproduce their algorithms in the short rebuttal phase.
>
> * **What are the limits of the approach? When would it fail?**
>
> Thank you for your valuable comments! One key limitation of our method is that the target task should be similar to the demonstration task, although they can be different. Therefore, if the difference between the new task and original task is too large, our method might fail. To quantize this, we extend our experiments to more significant difference situations. Specifically, we gradually increase the hole random range and compare the performance between our method and the strongest baseline. Results are shown in Figure 3 of the revised paper and Figure 7 in revised appendix. There is no performance improvement when the hole random range reaches 14mm.
>
> * **Would this work for pick-and-place or other robot tasks?**
>
> Our approach is generic which can be applied to any robotic learning tasks that receive sparse rewards and have access to demonstrations from a different but similar task. Due to the page and time limits, we only conduct the experiments on insertion tasks. In the future we will apply the proposed method to more other tasks.
>
> * **It would have been more helpful had the paper removed Figure 2 and spent more time describing the algorithm itself. The mismatched reward problem is relatively self-explanatory…... If a robot fails to accomplish a new task because it has learned a different one, it may not be able to receive any (sparse) rewards, so learning will be difficult.**
>
> We are sorry for the confusion caused by our statement, and based on your valuable comments, we have simplified the explanation of blocked reward signal issue in section 4.1 and removed Figure 2. You can intuitively understand it as following: imitating the original demonstrations in new tasks may lead to consistent failure and obtain no sparse rewards from the environment. Our method uses reward shaping to incorporate the information from the demonstrations to the sparse reward of the new task, instead of purely imitating the original demonstrations. By doing so, we can guide the learning process while keeping the policy optimal in the new task.
> In addition, we also provide detailed description of the algorithm and some theoretical analysis (in Section 4.2) to help better understanding. You can find details of the modifications (marked blue) in the revised paper. Thank you for your time!
>
> * **Prior work has also considered reward shaping which should be considered**
>
> Thanks for your insightful advice! Based on this suggestion, we implement the method in the paper: Shaping Rewards for Reinforcement Learning with Imperfect Demonstrations using Generative Models [https://arxiv.org/pdf/2011.01298.pdf]. This paper uses the output of a discriminator as the potential function. We added this approach as a baseline (named RS-GM) and performed additional experiments on several tasks and show their results in Figure 2 of the revised paper, where our method performs significantly better in most tasks.
>
>
>
> Thanks again for your time and insightful comments!

---

> > ### Comment · Reviewer_hEn1 · 2022-08-27
> > **Baselines**
> >
> > I appreciate the authors carefully considering this reviewer's feedback. In particular, adding the baseline "Shaping Rewards for Reinforcement Learning with Imperfect Demonstrations using Generative Models" is impactful.
> >
> > The addition of the sensitivity analysis ("Specifically, we gradually increase the hole random range and compare the performance between our method and the strongest baseline") is also strong.
> >
> > This reviewer is a little confused about the authors' statement that "both paper Brown et al., CoRL'20 and Chen et al. CoRL'21 do not release their code." A cursory Google search for each revealed the codebases below. Can the authors please let this reviewer know if the reviewer is mistaken or misunderstanding the authors' statement?
> >
> > https://github.com/dsbrown1331/CoRL2019-DREX
> >
> > https://github.com/CORE-Robotics-Lab/SSRR

---

> > > ### Author Response · Authors · 2022-08-28
> > > **Thanks for your time and efforts in reviewing our revised paper!**
> > >
> > > We really apologize that we missed the code link in the Brown et al. CoRL'19 paper. And as for Chen et al. CoRL'20, they did not mention their code in the original paper [https://arxiv.org/pdf/2010.11723.pdf]. And we searched paper Chen et al. CoRL'20 on CoRL official website [https://corlconf.github.io/corl2020/paper_281] and there is no code button there. We are sorry for our carelessness.
> > >
> > > Thanks for your time and efforts in reviewing our paper!

---

### Official Review · Reviewer_CiFM · 2022-08-01

**Originality:** Fair
**Technical Quality:** Good
**Clarity Of Presentation:** Good
**Impact:** 2

**Recommendation:**

Weak Reject: I recommend rejecting the paper, but will not argue for my recommendation if the majority of other reviewers have a different opinion.

**Summary:**

This paper proposes a learning from demonstration method to learn a certain task from demonstrations of a different task. Starting from a recent method that is able to learn from off-policy data using a replay buffer, they additionally include a shaped reward function to provide a learning signal even in failed trajectories. A key component of this shaped reward is explicitly setting actions that lead to out of distribution states to 0 reward. In their experiments, they show that their method achieves superior performance in two settings: (1) when environment dynamics differ (in their case changing the shape of a hole for an insertion task) and (2) when both environment dynamics and the reward differs (in their case randomly perturbing the location of the hole from the demonstrations). They also include a detailed ablation study.

**Issues:**

Section 4.1 and Figure 2: Rewrite/update to better explain phenomenon of blocked reward signal backprop.
Intro/Related Work: Discussion of inverse RL and it's limitations could be improved
Occasional typos

**Quality Of The Limitations Section:**

Additional details required

**Reviewer Expertise:**

4: The reviewer is confident but not absolutely certain that the evaluation is correct

**Robotics Focus:**

Sufficient demonstration on hardware

**Strengths And Weaknesses:**

Strengths
1. Rigorous experiments considering numerous baselines. The constructed baseline of extending SAC with the method of Vecerik et al. is a nice addition. Ablation studies illustrating the impact of model components is also good. Also show zero-shot sim-to-real transfer.
2. Methods section is clear, and the proposed method is simple and principled.

Weaknesses
1. While the related work is thorough, I am missing the motivation here. Why not just use inverse RL? The authors state that inverse RL methods are not able to learn from environment rewards, but this is a bit of a misrepresentation, as most inverse RL methods are assuming that such a reward function is unavailable.
2. Explanation of "reward signal backpropagation blockage" is confusing to me. Section could be reworked to better explain the phenomenon. The corresponding figure 2 is also confusing, not sure if it is adding to the understanding.
3. No baselines considering inverse RL methods.
4. Only one type of task (object insertion) is considered. What about multi-step tasks? Navigation? I feel that the method may be more limited when considering more significant shifts between demonstrated and target task as the reward shaping mechanism seems to rely on the original value function learning a complete enough representation to transfer to new tasks, which is a strong assumption.
5. No mention of how many demonstrations are used.

**Summary Of Recommendation:**

The experiments are well constructed with nice results, and the method is well-principled and elegant. However, as mentioned before, I do not completely see the motivation behind this method, and one of the main issues addressed, the blocked reward signal backpropagation, is unclear. Also, evaluation on only a block-insertion task is limited. If the authors can provide a better motivation and clarify the blocked reward signal section, I would be willing to change my score.

---

> ### Author Response · Authors · 2022-08-22
> **Response to Reviewer CiFM - Part(1/2)**
>
> Thank you so much for your insightful comments on our paper. Your review helps us a lot on revising our paper. We have uploaded the revised paper with modified places marked in blue, and we hope our following responses answer your concerns about our paper.
>
> * **Why not just use inverse RL? The authors state that inverse RL methods are not able to learn from environment rewards, but this is a bit of a misrepresentation, as most inverse RL methods are assuming that such a reward function is unavailable. No baselines considering inverse RL methods.**
>
> In our setting, we target on solving sparse reward robotic problems such as insertion etc. In such setting, there exist environment rewards, e.g, receiving a positive reward once successfully inserted in the hole. However, such reward is pretty sparse, making reinforcement learning very difficult. An effective way of solving the sparse reward RL problem is to guide learning using information from demonstration data, e.g, using IRL/IL methods. Therefore, we believe IRL/IL methods do not have to assume that the environment reward is unavailable. Furthermore, in our paper we consider settings where the target tasks are mismatched from the original tasks that provide the demonstrations data. We novelly use reward shaping to incorporate the information from the demonstrations to the sparse reward of the new task, which accelerates the training process of new tasks while keeping the optimal policy unchanged. As for baselines considering IRL methods, generally speaking, GAIL can be considered as a specific IRL method, where the learned rewards are surrogate rewards represented by a discriminator in a GAN framework. Furthermore, GAIL performs better than traditional IRL methods [1].
>
> * **Explanation of "reward signal backpropagation blockage" is confusing to me. Section could be reworked to better explain the phenomenon. The corresponding figure 2 is also confusing, not sure if it is adding to the understanding**
>
> We are sorry for the confusion caused by our statement, and based on your valuable comments, we have removed Figure 2 and rewritten this section (Section 4.1) to make it easier to understand. In addition, we also provide detailed description of the algorithm and some theoretical analysis (in Section 4.2) to help better understanding. You can find details of the modifications (marked blue) in the revised paper. Thank you for your time!
>
> * **I feel that the method may be more limited when considering more significant shifts between demonstrated and target task as the reward shaping mechanism seems to rely on the original value function learning a complete enough representation to transfer to new tasks**
>
> This is a very insightful comment. Let us first consider two extreme cases. In the first case, the expert environment is exactly the same as the target environment, this situation has been well studied with many mature imitation learning algorithms. Also considering the other extreme where the target tasks are completely unrelated to the original task, the demonstrations should not contain useful information for new tasks. However, most cases lie between these two extremes, e.g, the target task and the expert task have a certain similarity but are not identical. Our work attempts to fill the gap between the two extremes by reward shaping from demonstrations using an estimated value function.
> Moreover, we do not require the estimated value function to be completely precise in new tasks, they can just provide rough guidance to new tasks without affecting the policy optimality, which is much better than learning from scratch.
> To better explore the limitation of our method as you concern, we also extend our experiments to more significant shift situations. We gradually increase the hole random range and compare the performance between our method and the strongest baseline. Results are shown in Figure 3 of the revised paper and Figure 7 in revised appendix. Our proposed methods outperforms baseline until the task largely shift with 14mm*14mm random position.

---

> > ### Author Response · Authors · 2022-08-22
> > **Response to Reviewer CiFM - Part(2/2)**
> >
> >
> >
> > * **No mention of how many demonstrations are used.**
> >
> > We use 40 demonstration trails with 50 time steps each, and result in 2000 transitions in total. We have added this information in section 5.1 in the revised paper.
> >
> > * **If the authors can provide a better motivation and clarify the blocked reward signal section, I would be willing to change my score.**
> >
> > In many robotic learning problems, only sparse rewards are available, e.g. receiving a 0-1 reward when insertion is successful for the peg-in-hole task. In such setting, reinforcement learning can be very difficult and learning from demonstration can guide and significantly accelerate the learning process. However, existing LfD methods many focus on settings where the target task is same as the task that provides demonstrations. The motivation of this paper is to look in particular at domain shift between the target task and the demonstration task. This is a very important problem in robotic learning with demonstrations, since robotic tasks are usually pretty diverse (different shapes and positions of the holes for peg-in-hole tasks as considered in this paper), and collecting demonstrations for all tasks can be extremely expensive and inefficient. Therefore, the goal of this paper is to learn better policies for a variety of target tasks, when only demonstrations from a single different task are provided. This will allow us to handle new tasks better and faster without the need of collecting new demonstration data. To do this, we use reward shaping to incorporate the information from the demonstrations to the sparse reward of the new task, which accelerates the training process of new tasks while keeping the optimal policy unchanged. This is novel among approaches of learning with demonstrations from mismatched tasks, and performances better than baselines.
> >
> > Furthermore, we have simplified the explanation of blocked reward signal issue in section 4.1, and provided some theoretical analysis in section 4.2. You can intuitively understand it as following: imitating the original demonstrations in new tasks may lead to consistent failure and obtain no sparse rewards from the environment. Our method uses reward shaping to incorporate the information from the demonstrations to the sparse reward of the new task, instead of purely imitating the original demonstrations. By doing so, we can guide the learning process while keeping the policy optimal in the new task.
> >
> >
> >
> > Thanks again for your time and insightful comments!
> >
> > [1] Justin, Fu, et al. "Learning Robust Rewards with Adversarial Inverse Reinforcement Learning" NIPS2016

---

> ### Author Response · Authors · 2022-08-28
> **We are always ready to address any of your remained concern.**
>
> We deeply appreciate your time in reviewing our paper!
>
> As it comes to the end of the rebuttal phase,  we hope we have solved all your concerns and we are always ready to address any of your remained concerns.
>
> We deeply appreciate that if you could reconsider the score accordingly.
>
> Thanks again for your time and efforts in reviewing our paper!

---

### Official Review · Reviewer_rKEH · 2022-08-03

**Originality:** Good
**Technical Quality:** Good
**Clarity Of Presentation:** Very Good
**Impact:** 3

**Recommendation:**

Weak Accept: I recommend accepting the paper, but will not argue for my recommendation if the majority of other reviewers have a different opinion.

**Summary:**

This paper proposes a method to leverage information from expert traces on an MDP M to help solve a sparse reward task on an MDP M', where M' differs from M only by its reward.  The method involves performing policy evaluation on expert traces on M to calculate a value function V_M, and then leveraging that value function as a potential function for reward shaping on M'.  Experiments are performed both in simulation and on a physical robot for variants of an insertion task, where the sparse-reward variants of the task are shown to be solved faster than with existing methods thanks to the use of information from an expert solving a different instance of a similar task.

**Issues:**

 I'm not convinced the 'sparse reward signal backward propagation blockage' is exactly what's going on, or if it is that would require further discussion.
I'm not sure I understand the point of using pure imitation algorithms such as GAIL as baselines, as it seems pretty obvious they would not be able to solve a novel task.  Perhaps I missed something here in how the setup is done, but I'm not sure these provide much insight beyond BC, there is no expectation they would work in this setup anyways.
If the authors could convince me of the novelty of the contribution, and the types of scenarios where one would have access to a high-quality reward on the expert demonstrations, that would allow the calculation of V, as well as perhaps baselines that also use reward shaping coming from different potential functions (perhaps inspired from some of the existing literature) this could potentially make for a stronger paper.

**Quality Of The Limitations Section:**

Limitations are not well addressed

**Reviewer Expertise:**

3: The reviewer is fairly confident that the evaluation is correct

**Robotics Focus:**

Sufficient demonstration on hardware

**Strengths And Weaknesses:**

This paper is well written and easy to follow, and the ideas are easy to understand.  The experiments are convincing and clearly applicable to a robotics domain.

I think the paper could be improved with some more pertinent baselines, and a bit more detail on the actual algorithm.  I am also dissapointed that the robot experiments are just a sim2real transfer of the simulator-learnt policy, whereas the whole point of this type of algorithm is to be able to perform sufficiently efficient sparse reward learning such as to be able to train directly on hardware.  Another concern is that this approach requires a dense reward for the original expert demonstrations, which is something that is generally not available in a realistic scenario.

The paper is also missing some import related work:
Shaping Rewards for Reinforcement Learning with Imperfect Demonstrations using Generative Models [https://arxiv.org/pdf/2011.01298.pdf]
Reinforcement Learning from Demonstration through Shaping [https://openreview.net/forum?id=S145CVGuWB]
Deep Reward Shaping from Demonstrations [https://ieeexplore.ieee.org/abstract/document/7965896]


**Summary Of Recommendation:**

I think the ideas in the paper, although well-executed, is not particularly novel.  Using a potential function from expert demonstrations to shape the reward for a given task is something that has been looked at on a couple of occasions.  The novelty of the paper is to look in particular at domain shift, and leverage the existence of a well-defined reward during the expert demonstration,

---

> ### Author Response · Authors · 2022-08-22
> **Response to Reviewer rKEH - Part(1/2)**
>
> Thank you so much for your insightful comments on our paper. Your review helps us a lot on revising our paper. We have uploaded the revised paper with modified places marked in blue, and we hope our following responses answer your concerns about our paper.
>
> * **Approach requires a dense reward for the original expert demonstrations, which is something that is generally not available in a realistic scenario.**
>
> We are sorry for some confusing descriptions in the original text. In fact, the demonstrations we used are still annotated with sparse rewards. This sparse reward is easy to obtain in realistic robotic problems, e.g, for peg-in-hole tasks, adding a simple range sensor in the hole to detect whether the peg is successfully inserted. In this paper, we only use dense reward to train the expert policies which are then used to collect demonstration data. Nevertheless, the process of collecting demonstrations can be replaced by teleoperation or manually dragging the robot arm, without any reward design. Regardless of how the demonstrations are collected, the data is all annotated with sparse rewards which is compatible as the real robotic task environment. Thank you for your comments, we have added explanations (marked blue in Section 5.1) in the revised paper.
>
> * **The paper is not particularly novel. Using a potential function from expert demonstrations to shape the reward for a given task is something that has been looked at on a couple of occasions. The novelty of the paper is to look in particular at domain shift, and leverage the existence of a well-defined reward during the expert demonstration.**
>
> Indeed, using potential function as reward shaping for a given sparse reward task is not a new idea. However, as you also agreed, the novelty of the paper is to look in particular at domain shift. This is a very important problem in robotic learning with demonstrations, since robotic tasks are usually pretty diverse (different shapes and positions of the holes for peg-in-hole tasks as considered in this paper), and collecting demonstrations for all tasks can be extremely expensive and inefficient. Therefore, the goal of this paper is to learn better policies for a variety of target tasks, when only demonstrations from a single different task are provided. This will allow us to handle new tasks better and faster without the need of collecting new demonstration data. To do this, we use reward shaping to incorporate the information from the demonstrations to the sparse reward of the new task, which accelerates the training process of new tasks while keeping the optimal policy unchanged. This is novel among approaches of learning with demonstrations from mismatched tasks, and performances better than baselines. Furthermore, as explained in the previous question, all demonstrations are tagged with natural sparse reward which are easy to obtain, and the process of obtaining demonstrations can be replaced by teleoperation or manually dragging which do not require reward design.
>
> * **I think the paper could be improved with some more pertinent baselines; as well as perhaps baselines that also use reward shaping coming from different potential functions**
>
> Thanks for your insightful advice! Based on this suggestion, we implement the method in the paper: Shaping Rewards for Reinforcement Learning with Imperfect Demonstrations using Generative Models [https://arxiv.org/pdf/2011.01298.pdf]. This paper uses the output of a discriminator as the potential function. We added this approach as a baseline (named RS-GM) and performed additional experiments on several tasks and show their results in Figure 2 of the revised paper, where our method performs significantly better in most tasks.
>
> * **I'm not convinced the 'sparse reward signal backward propagation blockage' is exactly what's going on, or if it is that would require further discussion.**
>
> We are sorry for the confusion caused by our statement, and based on your valuable comments, we have removed Figure 2 and rewritten this section (Section 4.1) to make it easier to understand. In addition, we also provide detailed description of the algorithm and some theoretical analysis (in Section 4.2) to help better understanding. You can find details of the modifications (marked blue) in the revised paper. Thank you for your time!

---

> > ### Author Response · Authors · 2022-08-22
> > **Response to Reviewer rKEH - Part(2/2)**
> >
> > * **I'm not sure I understand the point of using pure imitation algorithms such as GAIL as baselines, as it seems pretty obvious they would not be able to solve a novel task. Perhaps I missed something here in how the setup is done, but I'm not sure these provide much insight beyond BC, there is no expectation they would work in this setup anyways.**
> >
> > Here we add GAIL as a baseline to mainly help us introduce and compare with another baseline GAIfO. GAIfO is a variant of GAIL that considers the dynamic mismatch between the target new task and the original task, which has the potential to work in our settings, although experiments show it also does not perform well.
> >
> > * **I am also disappointeded that the robot experiments are just a sim2real transfer of the simulator learnt policy,whereas the whole point of this type of algorithm is to be able to perform sufficiently efficient sparse reward learning such as to be able to train directly on hardware.**
> >
> > Thanks for comments! Yes our ultimate goal is to implement the entire algorithm pipeline directly on a real hardware. However, the key of this paper is to propose an algorithm framework that can accelerate learning of policies for diverse new tasks with sparse rewards, when only demonstrations of a single different task are provided. In this paper, we successfully evaluated the whole algorithm pipeline in simulation (e.g, demonstrations and learning are all in simulation environments). If we consider simulation and real world as basically environments with different domain features (e.g, contact, friction...), then the success of our algorithm in simulation should imply likely success if we implement the same algorithm pipeline in the real world. However, such training pipeline in the real world is beyond the scope of this paper, as it requires significant efforts to solve issues such as real world reward calculation, task setting, etc [1], which is nontrivial and worth a separate new paper. Therefore, we just deployed learned policy from simulation to the hardware to show its potential application on real robots, and leave training directly on hardware as future work, which is discussed in the limitation part of the revised paper.
> >
> >
> >
> > Thanks again for your time and insightful comments!
> >
> > [1] Zhu, Henry, et al. "The Ingredients of Real World Robotic Reinforcement Learning." ICLR 2020.

---

> ### Author Response · Authors · 2022-08-28
> **We are always ready to address any of your remained concern.**
>
> We deeply appreciate your time in reviewing our paper!
>
> As it comes to the end of the rebuttal phase,  we hope we have solved all your concerns and we are always ready to address any of your remained concerns.
>
> We deeply appreciate that if you could reconsider the score accordingly.
>
> Thanks again for your time and efforts in reviewing our paper!

---

### Author Response · Authors · 2022-08-22
**Revised PDF paper for Rebuttal**

We thank all reviewers for their insightful feedback and for helping us revise our paper!
We have revised our paper to address reviewers’ comments. The major changes in revised papers are highlighted in blue:

* Section 2: We modify the description of inverse RL to make it more clear. And we add the related work on reward shaping.
* Section 4: We removed Figure 2 (in the original paper) and **add more descriptions of our method with pseudo-code**.
* Section 4: We **add some theoretical analysis** to better support our motivation. Detailed proof can be found in appendix 7.5.

* Section 5: We **add a baseline method** using reward shaping: Shaping Rewards for Reinforcement Learning with Imperfect Demonstrations using Generative Models [https://arxiv.org/pdf/2011.01298.pdf], and update the results in Figure 2.

* Section 5: We **extend our experiments to more significant shift situations** and compared our methods with the strongest baseline. The results are shown in Figure 3 and appendix Figure 7 of the revised papers.

* Section 6: **The limitation section is specified in detail, and includes suitable scenarios and possible future works.**

* Appendix: We add detailed proof for our theorem and additional experiments on tasks with larger task shifts.

We temporarily attached the appendix to the main paper for your convenience.

We hope we have addressed all your concerns and questions. Please let us know if there are still any concerns.

---

### Meta-Review · Area_Chair_m9Gi · 2022-09-06

**Recommendation:** Accept (Poster)
**Confidence:** 4

**Metareview:**

This work proposes a learning from demonstration method that utilizes demonstrations from a task that is different from the target task to be learned via reinforcement learning. This removes the requirement of obtaining expert demonstrations for every new task the robot encounters. This is in an interesting problem setting and likely of relevance to the robotics community.  Specifically the authors propose a reward shaping algorithm for reinforcement learning with demonstrations from mismatched tasks. The work is evaluated in simulation and real-world experiments, including comparisons against state-of-the-art Learning from Demonstration (LfD) methods.

The authors have addressed most of the weaknesses outlined by the reviewers, and the reviewers agree that this work proposes a novel (incremental) algorithm, the manuscript is clear and evaluations are thorough (though the core contribution is evaluation in simulation only). Overall I recommend accept.

**Best Paper Nomination:**

No